# The Binding of Different Substrate Molecules at the Docking Site and the Active Site of γ-Secretase Can Trigger Toxic Events in Sporadic and Familial Alzheimer’s Disease

**DOI:** 10.3390/ijms24031835

**Published:** 2023-01-17

**Authors:** Željko M. Svedružić, Vesna Šendula Jengić, Lucija Ostojić

**Affiliations:** 1Laboratory for Biomolecular Structure and Function, Department of Biotechnology, University of Rijeka, 51000 Rijeka, Croatia; 2Laboratory for Medical Biochemistry, Psychiatric Hospital Rab, Kampor 224, 51280 Rab, Croatia; 3Department of Chemistry & Molecular Biology, Medicinaregatan 9 c, Box 462, University of Göthenburg, 40530 Göthenburg, Sweden

**Keywords:** proteinopathy, neurodegeneration, supramolecular organization, Alzheimer’s disease, amyloid

## Abstract

Pathogenic changes in γ-secretase activity, along with its response to different drugs, can be affected by changes in the saturation of γ-secretase with its substrate. We analyze the saturation of γ-secretase with its substrate using multiscale molecular dynamics studies. We found that an increase in the saturation of γ-secretase with its substrate could result in the parallel binding of different substrate molecules at the docking site and the active site. The C-terminal domain of the substrate bound at the docking site can interact with the most dynamic presenilin sites at the cytosolic end of the active site tunnel. Such interactions can inhibit the ongoing catalytic activity and increase the production of the longer, more hydrophobic, and more toxic Aβ proteins. Similar disruptions in dynamic presenilin structures can be observed with different drugs and disease-causing mutations. Both, C99-βCTF-APP substrate and its different Aβ products, can support the toxic aggregation. The aggregation depends on the substrate N-terminal domain. Thus, the C99-βCTF-APP substrate and β-secretase path can be more toxic than the C83-αCTF-APP substrate and α-secretase path. Nicastrin can control the toxic aggregation in the closed conformation. The binding of the C99-βCTF-APP substrate to γ-secretase can be controlled by substrate channeling between the nicastrin and β-secretase. We conclude that the presented two-substrate mechanism could explain the pathogenic changes in γ-secretase activity and Aβ metabolism in different sporadic and familial cases of Alzheimer’s disease. Future drug-development efforts should target different cellular mechanisms that regulate the optimal balance between γ-secretase activity and amyloid metabolism.

## 1. Introduction

Alzheimer’s disease is a slowly progressing and ultimately fatal neurodegenerative disorder [1,2]. Alzheimer’s disease stands out among other malignant diseases as imposing the greatest financial burden on healthcare providers in developed countries [1,3,4]. Impressive drug development efforts have been mostly centered on the metabolism of the last 99 amino acids of the amyloid precursor protein (C99-βCTF-APP) [3,4]. Based on strong genetic results, the most frequent therapeutic targets are two aspartic proteases: membrane-anchored β-secretase, and membrane-embedded γ-secretase [2,3,5]. A number of different compounds have been developed. Compounds with different structures, different binding sites, different mechanisms of action, and different pharmacological properties have shown very impressive nanomolar potency [1,4,5]. This impressive list of diverse and potent compounds has not produced the desired results, but it clearly shows that the present challenges extend beyond routine medicinal chemistry. It appears that we need to address some unique features in the enzymatic mechanisms of β-secretase and γ-secretase before we can develop successful drug design strategies [4,6,7,8,9,10,11,12,13,14].

Several pathogenic changes in Aβ production can be observed when γ-secretase is gradually saturated with its substrate [7,10,13,14]. Saturation can be a result of different mechanisms that lead to a decrease in the catalytic capacity of γ-secretase [7]. Changes in the saturation of γ-secretase with its substrate can also significantly affect how the enzyme responds to potential drugs [6,9,11,12]. The earliest age of onset can be observed with mutants that have the best chance to reach saturation at the lowest substrate loads [7]. The protective islandic A673T mutation in the APP substrate is the only mutation that leads to a decrease in γ-secretase’s saturation with its C99-βCTF-APP substrate [15]. Control of the saturation of γ-secretase with its substrate can be a key physiological process [16]. The underlying mechanisms are still not understood [16].

Studies of the enzymatic mechanisms of γ-secretase have provided some surprising and fascinating insights about the disease, but they remain incomplete [17]. Frequent problems include inconsistent conclusions and irreproducible results. Accurate mechanistic interpretation depends on well-defined quantitative analysis [9,11,12,13]. Quantitative analysis of complex enzyme activity depends on accurate mathematical modeling [18,19,20,21,22], which can be challenging for complex enzymes [9,12,20]. Fortunately, computational studies of molecular structures can greatly simplify and advance interpretations of enzyme activity studies [8,23,24,25,26,27,28,29].

We use advanced computational methods and γ-secretase structures to address some of the open questions in mechanistic studies of γ-secretase activity [7,9,10,11,13,30,31]. We found that γ-secretase can bind two different substrate molecules in parallel—one at the docking site and one at the active site [17,29,30]. The second substrate binds to the most dynamic sites in γ-secretase’s structure that can be affected by disease-causing FAD mutations and by different drugs.

The presented two-substrate mechanism can explain many of the pathogenic changes in γ-secretase activity at the molecular structural level. The presented molecular mechanism can be used for building correlations between different enzyme-based, cell-based, animal, and clinical studies of Alzheimer’s disease [3,5,7,9,10,32]. Such correlations are crucial for the development of effective early diagnostic tools and drug development strategies [4].

## 2. Results

### 2.1. Multiscale Molecular Dynamics (MD) Studies of Dimerization of C99-βCTF-APP Molecules in Cholesterol–Lipid Bilayers (Figure 1 and Figure 2)

C99-βCTF-APP molecules have highly dynamic structures that can be readily affected by the experimental conditions and can be difficult to measure [3,27,33,34,35,36,37]. We used multiscale MD studies to capture possible interactions between C99-βCTF-APP molecules in a cholesterol–lipid bilayer (Figure 1 and Figure 2) [38].

We started all MD studies by building a full-length C99-βCTF-APP structure from the available NMR conformers (PDB: 2LP1 [33]; see Materials and Methods). The soluble N-terminal and C-terminal ends can readily fold into compact structures even when calculations start with the fully extended structures (Appendix A). Changes in Ramachandran angles show that the folded structures have transient loop and β-sheet forms (Appendix A). The transient folded structures are a result of competing interactions between different amino acids and polar lipid heads (Appendix A). Protein–lipid interactions can explain how C99-βCTF-APP structures can be affected by the lipid composition [3,27,33,34,35,36,37]. The constant competition between attractive and repulsive interactions can lead to numerous dynamic structures that can be difficult to capture in measurements [33] and are easily affected by the experimental conditions [39]. The surface of the transmembrane helix can be easily covered (rigidified?) by cholesterol, as suggested previously (Figure 1B, [37]).

Lys29–Lys54 distances showed that the transmembrane (TM) helix constantly fluctuates between the two main conformations (Figure 1A,B, PDB: 2LP1 [33]). Similar to previous observations [27], we found that the TM helix can vary between a 45.45 Å long fully extended structure (Figure 1B) and the shortest 33.09 Å long structure (Figure 1A). The average distance in MD studies was 37.7 ± 2.5 Å (Figure 2D). The compact forms were slightly more dominant, constituting about 62% of the results. The conformers are mostly driven by the unusual hinge in the structure in the position of the Gly37Gly38Val39Val40 sequence (Appendix A [27,33]). Changes in the Ramachandran angles showed that the TM section has a predominantly α-helix structure (Appendix A). The OH groups on Thr44 and Thr49 are trapped in the hydrophobic environment (Figure 1 and Figure 2A–C) and must form hydrogen bonds with the adjacent peptide bonds.

When placed together, C99-βCTF-APP molecules gradually form dimers driven by the diffusion in the bilayer and complementary electric fields (Figure 2). We analyzed dimerization starting with two free C99-βCTF-APP molecules that were placed 10–30 Å apart and facing one another in different orientations (Appendix A). The two C99-βCTF-APP molecules formed many transient contacts before dynamic conformers became trapped in a compact dimer (Appendix A). The transient interactions represent transitions through several local energy minima before the two structures lock in a stable dimer (Appendix A). The calculations were extended to represent 20 µs of molecular time, which is well beyond the time that it takes for protein RMSD values to reach a plateau (8 µs). About 24.2% of the molecular surface area formed an interaction interface in the final complex (Figure 2B,C). The RMSF values for the individual amino acids [40] show that dimerization is mostly driven by the conformers in the polar parts of the substrate (Figure 2E and Appendix A). The changes in Lys29–Lys54 distances showed that the two C99-βCTF-APP molecules did not have identical structures when they formed the dimer, and that dimerization depends on combined contribution from the extended and compact structures (Figure 2D). The final dimer structure can be stabilized with as many as 11 H bonds (Figure 2F) and 94 Å^2^ of the total interaction surface (Figure 2C). The extracellular N-terminal domain forms more H bonds than the intracellular C-terminal domain (Appendix A). Changes in the number of H bonds as a function of time show a stepwise increase in the number of H bonds, as they reflect a sequence of structural changes that drive gradual complex buildup (Figure 2F).

In conclusion, C99-βCTF-APP molecules have sticky and highly dynamic structures that can be readily affected by the experimental conditions (Figure 1 and Figure 2, Appendix A) [39]. Thus, C99-βCTF-APP molecules exist in parallel in numerous dynamic conformations (Figure 1 and Figure 2), which can be difficult to distinguish in different measurements [3,27,33,34,35,36,37]. We show that multiscale MD studies can trace different competing interactions and different transient structures down to atomic details (Appendix A). Multiscale MD studies can be used to fill the gaps in measurements of the dynamic structures of C99-βCTF-APP molecules [3,27,33,34,35,36,37].

**Figure 1 ijms-24-01835-f001:**
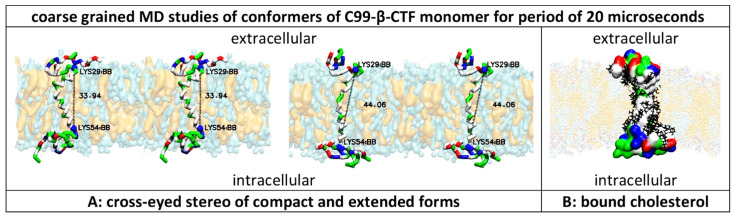
(**A**,**B**) Multiscale molecular dynamics studies of C99-β-CTF-APP’s structure in a cholesterol–lipid bilayer: Multiscale MD calculations can provide dynamic structural depictions of C99-β-CTF-APP’s structure that can be related to previous studies [27,33,37,41,42]. The amino acids are shown as hydrophobic (white), positive (blue), negative (red), and polar non-charged (green). The cholesterol–lipid bilayer shows surface models of cholesterol (orange) in a mixture of POPC, POPA, POPE, POPS, POPI, and PSM molecules (cyan) (Methods). (**A**) The transmembrane section of the C99-β-CTF-APP backbone can exist in compact and extended forms [27]. The transmembrane helix is hydrophobic (white), with notable polar sites at Thr 43 and Thr 48 (green) and hinge sites at Gly38–Gly39 (green). Changes in Lys29–Lys54 distances (numeration as in PDB:2LP1) show that the shortest conformer is around 33.94 Å long, while the longest conformer is about 44.06 Å long. About 62% of the time, the protein takes conformations that are about 37.7 ± 2.5 Å long. The extracellular and intracellular parts are rich in positive (blue), negative (red), and polar (green) amino acids. The extracellular and intracellular structures represent a dynamic network of competing interaction between different amino acids and polar lipid heads (Appendix A). (**B**) C99-β-CTF-APP’s Connolly surface shows that charged and polar amino acids in the extracellular and intracellular domains form compact structures atop the lipid bilayer [27]. The surface of the hydrophobic transmembrane section can be covered (rigidified?) with cholesterol molecules (black lines) [33].

**Figure 2 ijms-24-01835-f002:**
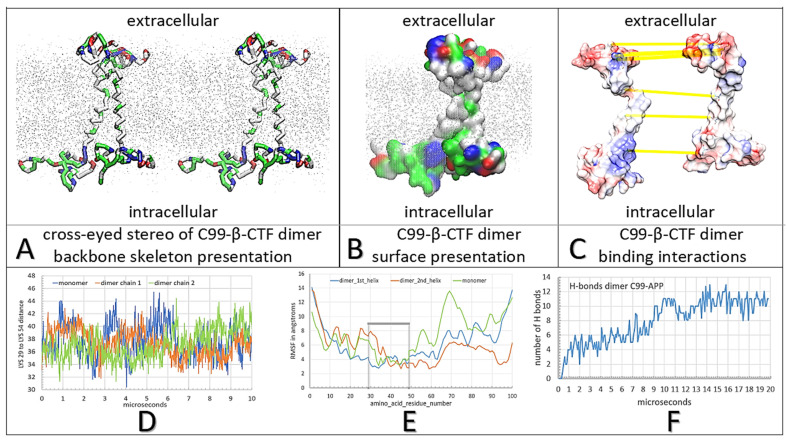
(**A**–**F**) Multiscale molecular dynamics studies of C99-β-CTF-APP dimerization in a cholesterol–lipid bilayer: The backbone models are used to show protein conformations, while the Connolly surfaces are used to show the size and shape of protein–protein contacts [43]. The amino acids are colored as hydrophobic (white), positive (blue), negative (red), and polar non-charged (green). For orientation, Thr43 and Thr48 sites are visible as green sites in the TM section. The cholesterol–lipid bilayer is shown as grey dots (methods). (**A**) Cross-eyed stereo view of the protein backbone used to illustrate conformational changes that support the formation of the C99-β-CTF-APP dimer (Appendix A). Dimerization is affected by conformational flexibility at Gly38 and Gly39 sites (green), which act as hinge points (Appendix A). The interaction between the soluble domains is a result of competing intramolecular and intermolecular interactions that form between charged and polar amino acids and lipid heads (Appendix A). (**B**) Surface models show that the two C99-β-CTF-APP molecules can wrap around one another to form large complementary surfaces down the full length of the protein. For clarity, the two molecules are shown as different shades of surface color (Appendix A). (**C**) The two molecules in the dimer are spread apart to show the complementary surface shapes and electric potentials (red–blue: −4.0 to 4.0 k_B_T/e). Highlighted are H bonds (yellow lines) [44]. The interaction takes place on the C-terminal domain between the positive Lys53-Lys54-Lys55 sites and negative Glu74-Glu75 sites. Highlighted on the N-terminal domain are interaction sites between Glu4 and Lys16, and between Arg5 and Glu22-Asp23 (Appendix A). (**D**) The changes in Lys29–Lys54 (numbering as in PDB: 2LP1) distances as a function of the MD calculation steps show that dimerization depends on the combined contribution from the extended and compact forms of the substrate [27]. (**E**) Relative differences in RMSF values as a function of amino acid position show that dimerization is mostly dependent on the conformers in the polar parts of the substrate [40,45]. (**F**) The rate of dimerization can be described by the number of H bonds formed between the two proteins as a function of the calculated molecular time (Appendix A). The initial lag represents the initial diffusion in the lipid bilayer before the first contact between molecules. The steps in the graph correspond to different conformers and local energy minima as the two structures form the most stable complex (Appendix A).

### 2.2. Multiscale MD Studies of Saturation of γ-Secretase with its C99-βCTF-APP Substrate (Figure 3)

We analyzed the extent to which the interactions observed between two free C99-βCTF-APP molecules (Figure 2) could be observed when one molecule was bound to γ-secretase as a substrate (Figure 3A–C). Most notably, a free C99-βCTF-APP molecule was used to challenge γ-secretase while the enzyme was processing its Aβ substrate (Appendix A). γ-Secretase can be simultaneously exposed to two substrate molecules when the enzyme is gradually exposed to increasing levels of its substrate [10,12,18,19,20,21], i.e., when the C99-βCTF-APP substrate starts to accumulate next to γ-secretase while the enzyme is still processing its substrate (Appendix A). Some studies indicate that γ-secretase has a separate substrate docking site and active site [9,30,46], or even that γ-secretase can bind multiple substrate molecules in parallel [9,10,12].

The parts of the C99 structures that support formation of C99 dimers (Figure 2) are not visible in the cryo-EM structures of γ-secretase [46] (see Materials and Methods). Thus, these structures are highly mobile and take multiple conformations, even when the substrate is bound and covalently fixed to the γ-secretase [46]. Possible conformers can be depicted by multiscale molecular dynamics studies [47]. We started MD studies with the Aβ substrate buried under the nicastrin ectodomain (Figure 3A–C). We started by first looking at γ-secretase in complex with the Aβ 1–49 substrate, which could be one of the key steps in pathogenic changes in Aβ production [13,28,48].

Coarse-grained MD studies started with free C99-βCTF-APP substrate facing γ-secretase with Aβ 1–49 bound in the active site (Figure 3A–C). The calculations were repeated with C99-βCTF-APP substrate placed in different orientations 10 to 30 Å apart from γ-secretase (Methods). The free C99-βCTF-APP substrate can diffuse in the bilayer and form contacts with γ-secretase driven by complementary electric fields (Appendix A). In all calculations, we found that the nicastrin ectodomain can gradually close over the N-terminal domain of the bound substrate (closure takes about 2 µs at the molecular time scale; Appendix A). The closure of the nicastrin ectodomain can compete with the formation of the first contacts between the free C99-βCTF-APP substrate and the nicastrin ectodomain (Appendix A). This closure was described in previous studies [24,25,26], all of which suggest that the closure has several regulatory functions. We found that the closure was driven by interactions between the nicastrin, presenilin 1, and presenilin enhancer 2 subunits (described in atomic detail in Appendix A).

Interestingly, we found in all calculations that the nicastrin ectodomain in its closed position can also affect the free substrate in reaching the presenilin 1 subunit (Figure 3D,E). The first contacts always form between the N-terminal domain of the free substrate and the nicastrin ectodomain domain (Appendix A). Initial interactions between nicastrin and the substrate have been suggested in previous studies [25]. Here, we go a step further, proposing that the initial contacts between the N-terminal domain of the substrate and the nicastrin ectodomain can control transient contacts between the C-terminal domain of the substrate and TM2 and TM3 on presenilin 1 (Appendix A). Such interactions could affect dynamic catalytic processes in the active site tunnel and the differences between the Aβ x-49 and Aβ x-48 product paths (Appendix A).

We further explored possible functions of the nicastrin ectodomain by repeating the CG-MD calculations with the nicastrin head fixed in its open position (Figure 3G–I, option posrestrain 1000 pN [47])). The restrained nicastrin did not close in calculations that represented as much as 20 µs of molecular events (Figure 3G,H). When the nicastrin ectodomain is open, the free substrate can immediately form contacts with the Aβ substrate and presenilin (Figure 3I). In the first contact, we observed as many as six hydrophobic interactions and up to five polar interactions (Figure 3G–I). Such interactions can readily affect the dynamic structures that control catalytic functions around TM2, TM3, TM6a, and the Aβ substrate [28,46]. The regulatory function of the nicastrin ectodomain was further demonstrated by repeating the multiscale MD studies with γ-secretase without nicastrin (Figure 3J–L). In the absence of interference by nicastrin, the free C99-βCTF-APP substrate can fully interact with presenilin 1 and the bound substrate (Figure 3J–L).

In conclusion, we present a novel two-substrate mechanism that can be viewed as an extension of the earlier structural studies [24,25,26,28,29,46]. We give a new significance to the earlier proposals that γ-secretase has a separate substrate docking site and active site [9,17,29,30,46]. We propose that γ-secretase can bind the second substrate molecule at its docking site while it is still processing its initial Aβ substrate (Figure 3 and Appendix A). The second substrate can bind to the most dynamic parts in the catalytic complex [46]. The same dynamic protein parts can be affected by disease-causing mutations and by binding of different drugs ([49]; Appendix A). These dynamic structures can also control the steps in processive catalysis—most notably the pathogenic differences between the Aβ x-49 and Aβ x-48 production paths [8,23,28,46]. Thus, the presented two-substrate mechanism can be used to analyze pathogenic changes in different situations that have a mismatch between the catalytic capacity of γ-secretase and its substrate load [10,11,13,14]. The presented two-substrate mechanism indicates that the nicastrin subunits can be the first step in the control of pathogenic interactions between the N-terminal domains of different fragments of amyloid molecules [24,25,26].

**Figure 3 ijms-24-01835-f003:**
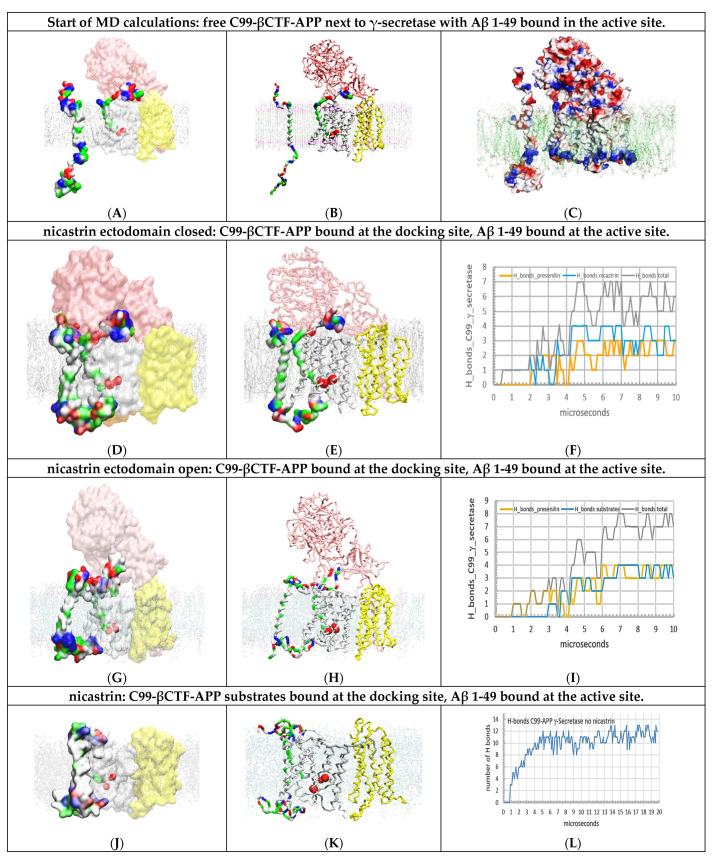
(**A**–**L**) Multiscale MD studies of docking of the free C99-βCTF-APP substrate to γ-secretase: Aβ 1-49 substrate buried under the nicastrin ectodomain. The γ-secretase complex (PDB:6IYC, [46]) shows nicastrin (pink), presenilin 1 (white), and Aph1 and Pen2 (yellow) subunits. Red beads depict the active sites Asp257 and Asp 385. The cholesterol–lipid bilayer is shown as dots (Materials and Methods). Surfaces of free C99-βCTF-APP (PDB:2LP1 [33]) and bound Aβ 1-49 substrates (PDB:6IYC, [46]) are colored as hydrophobic (white), negative (red), positive (blue), and polar non-charged (green). The backbone models are used to show protein conformers, while the partially transparent Connolly surfaces are used to show the size and shape of protein–protein contact surfaces [43]. (**A**,**B**) the starting structures for MD calculations depict a mismatch between the catalytic capacity of γ-secretase and its load of C99-βCTF-APP substrate. Increase in the C99-βCTF-APP substrate load leads to an increase in the chances that the free C99-βCTF-APP substrate can challenge γ-secretase while the enzyme is still processing its Aβ 1-49 substrate [10,12,18,19,20,21]. Thus, all MD calculations started with free C99-βCTF-APP substrate placed in different rotations 5 to 30 Å away from the γ-secretase-(Aβ 1-49) complex (Appendix A). (**C**) Connolly surfaces: red = negative, blue = positive, and white = not charged [44]. The electrostatic patches on the protein surfaces show that very specific conformational changes must form in MD studies to support the buildup of complementary docking interactions between γ-secretase and its C99-βCTF-APP substrate (Appendix A). (**D**–**F**) The first contact is observed between the N-terminal domain of free C99-βCTF-APP and the nicastrin ectodomain, as indicated in previous studies [24,25,26]. In the closed position, the nicastrin ectodomain can affect access of the C-terminal domain of the free C99-βCTF-APP substrate to presenilin 1 and the Aβ 1-49 substrate in the active site. The figure shows initial transient contacts between the C-terminal domain and the TM2, TM3, and TM6a sites in presenilin that can be observed with some conformers. (**G**,**I**) The mobility of the nicastrin ectodomain can be restricted in MD studies in an open position (command: POSRES @1000 pN, methods) [47]. With the nicastrin head open, the free C99-βCTF-APP substrate can dock with its full length to presenilin in the first contact. The C99-βCTF-APP bound at the docking site can form contacts with the N-terminal domain of the bound Aβ 1-49 substrate. The structures show how the nicastrin ectodomain can control the contacts between the two substrates [24,25,26]. (**J**–**L**) When the nicastrin subunit is removed, the N-terminal domain of the free C99-βCTF-APP substrate can have unimpeded interactions with the N-terminal domain of the bound Aβ 1-49 substrate. The rest of the free C99-βCTF-APP substrate forms tight contacts with presenilin—most notably the catalytic loops. The steps in the buildup of interactions can be described quantitatively by following the H bonds as a function of the calculated molecular time (panel **L**). The initial lag represents free diffusion and the first contact, while the stepwise changes in the number of H bonds represent gradual conformational changes in the buildup of interactions. The large interaction surface results in the highest number of H bonds, while the compact complex structure makes the two substrates merge.

### 2.3. Multiscale MD Studies of Nicastrin’s Function in the γ-Secretase Complex with the Exposed N-Terminal end of the Bound Aβ Substrate (Figure 4)

When an Aβ substrate is bound at the active site of γ-secretase, its highly mobile N-terminal domain can be hidden to various degrees under the nicastrin ectodomain (Figure 3 [46]). In one extreme, the N-terminal domain can be fully exposed at the external surface of the nicastrin ectodomain (Figure 4A). We prepared γ-secretase in a complex with the Aβ 1-49 substrate with its N-terminal domain exposed on the external surface of nicastrin (Figure 4A, Materials and Methods). The complex was challenged with free C99-βCTF-APP substrate. The aim was to analyze the extent to which the nicastrin ectodomain can prevent potentially toxic aggregation between the N-terminal domains of the two substrates (i.e., to compare the mechanisms in Figure 3 and Figure 4).

Multiscale MD studies show that the N-terminal domain of the substrate is highly flexible and always in contact with the nicastrin surface (Figure 4A). Different conformers always lead to some binding interactions because both proteins share numerous polar and charged groups on their surface (Appendix A). The N-terminal domain of the bound substrate cannot prevent the closure of the nicastrin ectodomain, but it can affect the related conformational changes (Figure 4C,D).

The docking of the free C99-βCTF-APP substrate is not significantly affected by the position of the N-terminal domain of the bound substrate (compare Figure 4 and Figure 3E,F). Most notably, the nicastrin ectodomain can bind the N-terminal domains of both substrates (Appendix A). Competition with nicastrin could control potentially toxic aggregation between two substrate molecules (compare Figure 4B with Figure 3D–F). We propose that the nicastrin ectodomain can prevent toxic aggregation between the N-terminal domains of the two substrates by different mechanisms in different conformations (compare Figure 3 and Figure 4). The closed nicastrin ectodomain can take multiple conformations in its function (Figure 4C,D). Different conformers can explain why cryo-EM studies could not capture nicastrin’s structure in its different closed conformations [46].

**Figure 4 ijms-24-01835-f004:**
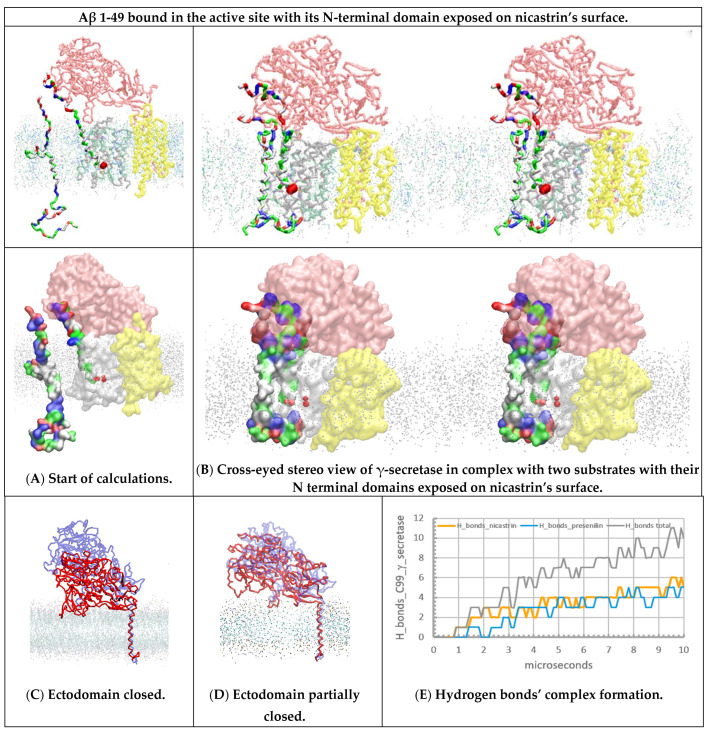
(**A**–**E**) Multiscale MD studies of the docking of the free C99-βCTF-APP substrate to γ-secretase: Aβ 1-49 substrate exposed on the external nicastrin surface. The γ-secretase complex (PDB:6IYC, [46]) shows nicastrin (pink), presenilin 1 (white), and Aph1 and Pen2 (yellow) subunits. Red beads depict the active sites Asp 257 and Asp 385. The cholesterol–lipid bilayer is shown as dots (Materials and Methods). Surfaces of the free C99-βCTF-APP (PDB:2LP1 [33]) and bound Aβ 1-49 substrates (PDB:6IYC, [46]) are colored as hydrophobic (white), negative (red), positive (blue), and polar non-charged (green). The backbone models are used to show protein conformers, while the partially transparent Connolly surfaces are used to show the size of the protein–protein contact surfaces [43]. (**A**) All calculations started with the free C99-βCTF-APP substrate positioned between 5 and 30 Å away from γ-secretase in complex with the Aβ 1-49 substrate. Just as in Figure 3A,B, the nicastrin ectodomain is open at the start of the calculation [24,25,26], but this time, the N-terminal domain of the bound Aβ 1-49 substrate is fully exposed on the external nicastrin surface. (**B**) Free C99-βCTF-APP can diffuse through the membrane and dock with its full length to the γ-secretase-(Aβ 1-49) complex. First, the N-terminal parts of the two substrates compete in interactions with the nicastrin ectodomain (Appendix A). Second, the TM parts of the two substrates form binding interactions at the start of the active site tunnel (sites between TM2 and TM3). Third, the C-terminal part of the free C99-βCTF-APP forms transient interactions with the most dynamic presenilin sites—the cytosolic end of the active site tunnel TM2, TM3, TM6, TM6a, and TM7 [28,46]. (**C**,**D**) The nicastrin ectodomain at the start (blue) and the end (red) of the MD calculations. The models on the right show that the exposed N-terminal domain of the Aβ 1-49 substrate can affect the closure of the nicastrin ectodomain. The closed ectodomain can form many conformers, which can be difficult to capture by cryo-EM studies [46]. (**E**) The rate of interaction buildup can be described by following the number of H bonds as a function of the calculated molecular time. The lag time represents the initial diffusion and the first contact, while the stepwise changes in the number of H bonds represent gradual conformational changes in the buildup of the complex.

### 2.4. Multiscale Molecular Dynamics Studies of γ-Secretase with Two Substrates of Different Lengths (Figure 5)

γ-Secretase can use substrates of different lengths—most notably, substrates that come from the α-secretase and β-secretase reaction paths: C83-αCTF-APP and C99-βCTF-APP, respectively [46]. The substrate length can affect the pathogenic changes in Aβ metabolism [15]. We used multiscale MD studies to analyze interactions between γ-secretase and the substrates of different lengths (Figure 5). The transmembrane section of the free substrate (starting Val12-His13-His13-Gln15; ending with Lys53-Lys54-Lys55-Gln56-Tyr57) was used to challenge γ-secretase in complex with Aβ catalytic intermediates with the shortest N-terminal domains (starting at Val17) [46].

We found that, in all cases, two substrate molecules can bind simultaneously to γ-secretase (Figure 5); however, the shorter substrates can lead to a smaller number of binding interactions (compare Figure 3, Figure 4 and Figure 5). Thus, the docking of the shorter free substrate is less likely to affect ongoing γ-secretase activity. These results are consistent with those of other studies showing that shortest substrates are less likely to support pathogenic changes in Aβ products [15].

**Figure 5 ijms-24-01835-f005:**
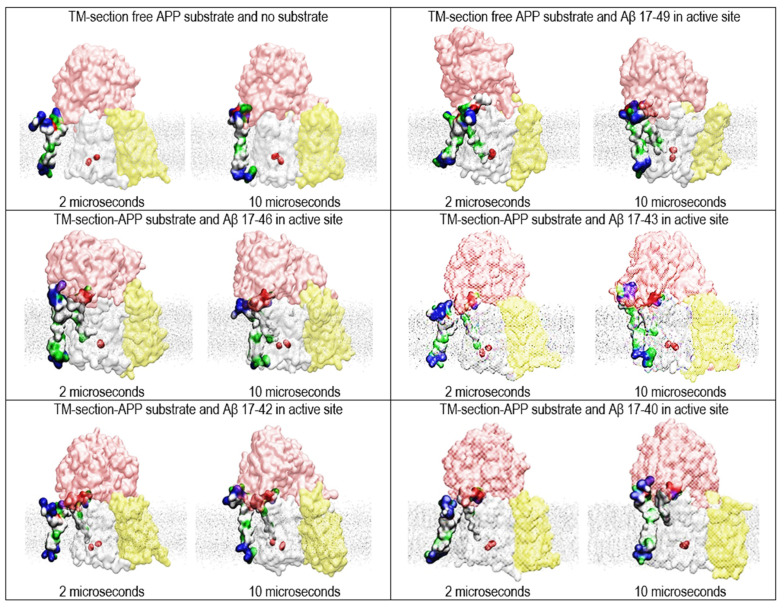
Multiscale MD studies of the docking of the short forms of free substrates to γ-secretase: substrate fragments bound in the active site (PDB:6IYC [46]). The γ-Secretase complex is depicted as partially transparent Connolly surfaces, showing nicastrin (pink), presenilin 1 (white), Aph1 (yellow), and Pen2 (not visible) [43]. Red beads depict the active sites Asp 257 and Asp 385. The surfaces of the two substrates are shown as hydrophobic (white), negative (red), positive (blue), and polar non-charged (green). The cholesterol–lipid bilayer is shown as silver dots. The calculations used substrate forms of different length to show how substrate length can affect the presented two-substrate mechanism. The free substrate shows the NMR structure, starting with V12-H13-H13-Q15 and ending with 53K-54K-55K-Q56-57Y (PDB:2LP1 [33]). The substrates bound in the active site tunnel had different lengths of Aβ 17-x, representing different catalytic intermediates that can come from the 83-α-CTF-APP substrate [11,13]. The short N-terminal domain of the bound substrate is hidden under the nicastrin ectodomain [46]. We compared the structures at the first contact between the free substrate and γ-secretase (at 2 µs) with the structures of the fully formed complex (at 10 µs). The figures show that even the shortest substrates can form contacts that can affect dynamic changes in presenilin’s structure that drive processive catalysis [13].

### 2.5. Multiscale Molecular Dynamics Studies of the Docking of Free C99-βCTF-APP Substrate to γ-Secretase Complexes with No Bound Substrate (Figure 6)

Free C99-βCTF-APP substrate was used to challenge the γ-secretase complex with no bound substrate (Figure 6). In the absence of the bound substrate, the nicastrin ectodomain can close over presenilin 1 with no interference (Appendix A). In these conditions, the free C99-βCTF-APP substrate can dock to the nicastrin ectodomain, but it cannot reach the presenilin (Figure 6B–D).

In total, we have presented our two-substrate mechanism in four very different situations (Figure 3, Figure 4, Figure 5 and Figure 6): first, with the bound substrate fully buried under the nicastrin ectodomain (Figure 3 and Appendix A); second, with the bound substrate exposed on the surface of the nicastrin ectodomain (Figure 4); third, with substrates of different lengths (Figure 5); and fourth, with no substrates bound at the active site tunnel (Figure 6). These are just some of the selected interactions, out of many intermediate situations that we could explore in future drug development efforts (Appendix A). Different interactions show differences in the rate of contact buildup, in the contact sites, the size of the contact surface, and in the final number of H bonds observed (Figure 3, Figure 4, Figure 5 and Figure 6).

In parallel to some differences, all of the four presented interactions support several major conclusions. In all cases, the N-terminal domain of the docked substrate makes contact with the nicastrin ectodomain first (Figure 3, Figure 4, Figure 5 and Figure 6) [25]. These contacts can affect the substrate docking to various extents—most notably the contact between the C-terminal domain of the docked substrate and presenilin 1 (Figure 3, Figure 4, Figure 5 and Figure 6, Appendix A). Those interactions target the sites that can be affected by FAD mutations and drugs (Appendix A). Thus, future drug design efforts could control contacts between the docked C99-βCTF-APP substrate and presenilin by targeting the closure and opening of the nicastrin ectodomain (Appendix A).

**Figure 6 ijms-24-01835-f006:**
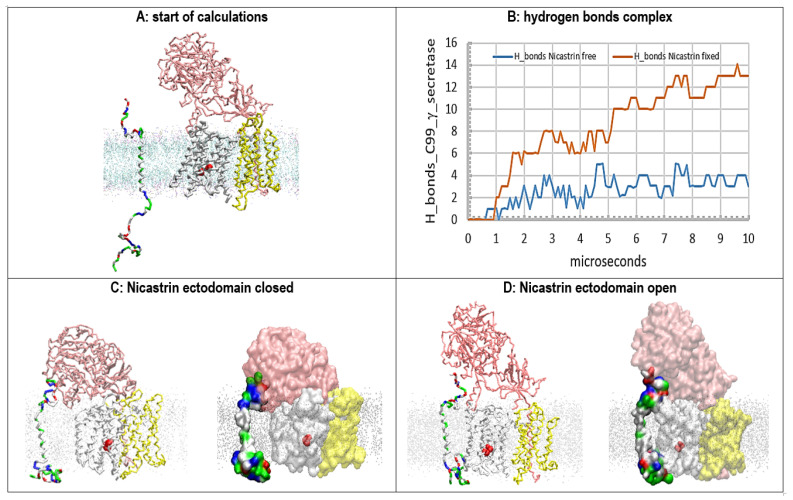
(**A**–**D**) Multiscale MD studies of the docking of the free C99-βCTF-APP substrate to γ-secretase: no substrate bound in the active site. The γ-secretase complex shows nicastrin (pink), presenilin 1 (white), and Aph1 and Pen2 (yellow). Red beads depict the active sites Asp 257 and Asp 385. The silver dots represent the cholesterol–lipid bilayer. The free C99-βCTF-APP substrate is colored as hydrophobic (white), negative (red), positive (blue), and polar non-charged (green). The backbone models are used to show protein conformers, while the partially transparent Connolly surfaces are used to show protein–protein contacts [43]. (**A**) Residue-based coarse-grained MD calculations started with the fully extended C99-β-CTF-APP structure that was positioned between 5 and 20 Å away from the γ-secretase complex with its nicastrin ectodomain open. (**B**) The buildup of protein–protein interactions is described quantitatively by counting H bonds that form in calculated molecular time (10 µs). The initial lag represents the diffusion time and the conformational changes that take place after the first contact. The subsequent steps in the graph represent conformational changes that drive the buildup of binding interactions. (**C**) In the closed position, the nicastrin ectodomain can interfere with the access of the free C99-βCTF-APP substrate to the TM2 and TM3 sites in presenilin 1. A maximum of 3–4 H bonds is observed when the free nicastrin head falls and blocks the substrate from reaching the presenilin. (**D**) The mobility of the nicastrin ectodomain can be restricted in MD studies in an open position (POSRES @1000 pN) [47]. With the nicastrin head open, the free C99-βCTF-APP substrate can dock with its full length to presenilin 1 and nicastrin. The docking sites overlap with dynamic presenilin structures that control the processive proteolytic cleavages in Aβ production [46]. For some of the conformers, as many as 13 hydrogen bonds can be observed when the substrate is docked over the full length of the γ-secretase complex.

### 2.6. AA-MD Studies of Docking of the C-Terminal Domain of C99-βCTF-APP to the Cytosolic Section of the Presenilin Subunit (Figure 7)

We have shown thus far that the docking of the N-terminal domain of the free C99-βCTF-APP substrate to nicastrin can lead to gradual docking of its C-terminal domain to the presenilin subunit (Figure 3, Figure 4 and Figure 5, Appendix A) [24,25,26]. The C-terminal domain docks to the most dynamic parts in the presenilin structure that can control processive catalysis [28,46]. Thus, we analyzed docking to presenilin at the atomic level, using conversion from coarse-grained to all-atom structures (Figure 7 and Appendix A). The coarse-grained structures that represent the first contacts between the C-terminal domain of the free substrate and the cytosolic end of presenilin 1 were used to prepare the all-atom structures (Appendix A [50]). Different calculations show some differences in the docking sites (Appendix A), and the related structural changes in the active site tunnel on the presenilin subunit (Appendix A [28]). The differences can be attributed to different contact surface sizes and different docking orientations. Similar effects can be caused by changes in protonation of the two Asp residues in active sites, or by changes in lipid composition in the membrane (Appendix A).

However, we also found some common features in all of our docking calculations (Figure 7). In all calculations, the docking of the substrate’s C-terminal domain gradually spread apart the cytosolic ends of TM2, TM3, and TM6 on the presenilin subunit by acting on the connecting structural loops and TM6a (Figure 7A and Appendix A). The spreading resulted in the opening of the active site tunnel and an increase in the distance and angle between the active sites Asp257 and Asp385 (Figure 7B,C). The docking affects the presenilin structure at the sites that are most frequently affected by FAD mutations (Appendix A). The docking also affects known drug-binding sites [8,23]. In sum, docking of the second substrate can increase the average distance between the active sites Asp257 and Asp385, just like the FAD mutations (Appendix A), the binding of drugs [8], or a switch from a POPC bilayer to a mixed cholesterol–lipid bilayer (Appendix A).

We propose that docking of the C-terminal domain of the C99-βCTF-APP substrate to the cytosolic end of the presenilin subunit could explain how saturation with its substrate leads to changes in γ-secretase activity [6,9,10,13,14,48]—specifically, the shifts from Aβ(x-49) to Aβ(x-48) production [10,13,48], the increase in the Aβ(x-42)/Aβ (x-40) ratio [13,48], and changes in the enzyme’s response to different drugs [6,9].

**Figure 7 ijms-24-01835-f007:**
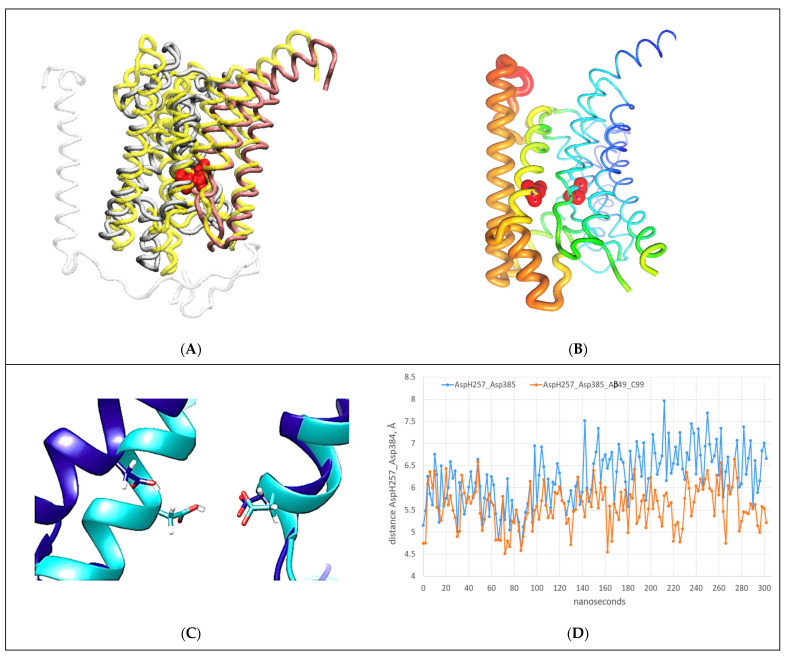
(**A**–**D**) All-atom MD studies of docking interactions between the C-terminal section of the C99-βCTF-APP substrate and the cytosolic section of the γ-secretase-(Aβ 1-49) complex [46]: (**A**) Presenilin’s structure (gray) with C99-βCTF-APP fully docked (white) is superimposed on presenilin’s structure before the docking (N-terminal domain = yellow, C-terminal domain = pink) [46]. The superimposed structures show that C99-βCTF-APP bound at the docking site can spread apart the cytosolic ends of TM2, TM3, TM6, and TM6a by acting on the loops between the TM regions (Appendix A). This spreading can affect key parts in the processive catalysis [28], including the active sites AspH257 and Asp385 (shown in red). For clarity, the figure depicts only the parts that participate in the interactions (Figure 3D and Appendix A). (**B**) The changes in mobility of different presenilin parts that can be observed during substrate binding at the docking site are illustrated using principal component analysis and Bio3D protocols [45]. Differences in mobility are illustrated using a rainbow scale: the thin blue lines represent the lowest mobility, while the thick red lines represent the highest mobility. For orientation, the active sites AspH257 and Asp385 are shown as red beads. The docking predominantly affects the presenilin structure at the sites that can be affected by drugs [8,23] and FAD mutations (https://www.alzforum.org/mutations accessed on 15 December 2022). (**C**) The active structures superimposed before (cyan) and after (blue) full docking of free C99-βCTF-APP to the cytosolic end of the presenilin subunit. The docking leads to increases in the angle and distance between the active sites Asp257 (protonated) and Asp385 (unprotonated). Such changes can affect the optimal catalytic structures of γ-secretase [28]. (**D**) AA-MD analysis of changes in distance between γ-carbon atoms on AspH257 and Asp385 caused by C99-βCTF-APP docking as a function of the calculated molecular time. We compared the changes in AspH257 and Asp385 distances caused by the docking (blue) with the changes in the absence of C99-βCTF-APP (red). C99-βCTF-APP docking leads to a wider active site structure, in the same way as FAD mutations (Appendix A) and drugs that target the active site tunnel [8,23].

### 2.7. Substrate Channeling between BACE1 and γ-Secretase (Figure 8)

The results presented here suggest that the nicastrin ectodomain can play key functions in controlling the binding of the second substrate to catalytically active γ-secretase (Figure 3, Figure 4, Figure 5 and Figure 6). In this respect, the presented two-substrate mechanism is an extension of previous studies [24,25,26]. In cells, the opening and closing of the nicastrin ectodomain could be regulated by a supramolecular complex between γ-secretase and β-secretase (BACE1) [51]. Such interactions would indicate that substrate channeling could regulate substrate docking to γ-secretase [52] and, thus, disease pathogenesis and pharmacology [53]. Regulation of enzyme activity by substrate channeling is frequently observed in metabolic studies [52]. Proteins in cells are present in exceptionally high concentrations that favor substrate channeling and supramolecular interaction [52]. We used multiscale MD studies to analyze possible docking interactions between β-secretase and the ectodomain of nicastrin (Figure 8).

The soluble part of the β-secretase structure was positioned facing the nicastrin ectodomain in the plane of presenilin 1 (Appendix A). The two proteins can rapidly form transient interactions with their polar surfaces (Appendix A). The specificity of the presented docking interactions was tested by conducting a series of docking studies with β-secretase placed at different orientations and distances. β-Secretase was placed 5 to 15 Å away from nicastrin to analyze how initial contacts can compete with the closure of the nicastrin ectodomain (Figure 8E). Different orientations were used to study how contact sites can affect the rate of complex formation and the related conformational changes (Figure 8B). We found that the structures of both proteins were highly flexible and readily affected by the contacts (Appendix A). We found conformational changes that support large interaction surfaces with matching surface potentials and shapes (Appendix A). It is very important to note that the docking studies intentionally used the soluble part of β-secretase’s structure that is not anchored to the membrane by its transmembrane helix (PDB:4FGX, [54]). Thus, the soluble structure can readily diffuse to the surrounding solvent, unless it forms binding interactions with γ-secretase (Appendix A). Docking studies showed that β-secretase always forms contacts with the nicastrin ectodomain with its C-terminal-facing membrane surface (Figure 8A, black spheres). This is the expected position for the transmembrane domain for the β-secretase structure [54], which could support the significance of the presented docking orientation.

Repeated calculations showed that the biggest interaction surface was observed when the nicastrin ectodomain was open and facing β-secretase in a plane that is parallel to TM2 and TM3 on presenilin 1 (Figure 8A,B). Such interactions can induce structural changes in both β-secretase and γ-secretase that can facilitate the buildup of large interaction surfaces (Appendix A). β-Secretase can embrace nicastrin by opening its active-site loops (Figure 8D)—specifically residues 78–86, 113–115, 317–322, and 372–379 (Appendix A) [54]. The nicastrin ectodomain will mold to β-secretase’s structure with its highly dynamic β-sheet structures (Figure 8B). Interaction domains are positioned between residues 310–319, residues 499–512, and residues 577–602 [51] (Appendix A).

Interaction between BACE1 and γ-secretase can regulate the opening and closing of the substrate’s docking site (Appendix A). Interaction between BACE1 and nicastrin pushes the nicastrin ectodomain in the opposite direction from the conformational changes that take place when the ectodomain is guarding the substrate’s access to the docking site (compare Figure 8 with Figure 3, Figure 4, Figure 5 and Figure 6). The complex between BACE1 and γ-secretase also stretches the long and flexible loop between TM2 and TM1, thereby opening the space between TM2 and TM3 (compare Figure 8B with Figure 3C,D). Closing and opening of the substrate’s binding sites is known to regulate transient protein–protein interactions in the case of substrate channeling [52].

**Figure 8 ijms-24-01835-f008:**
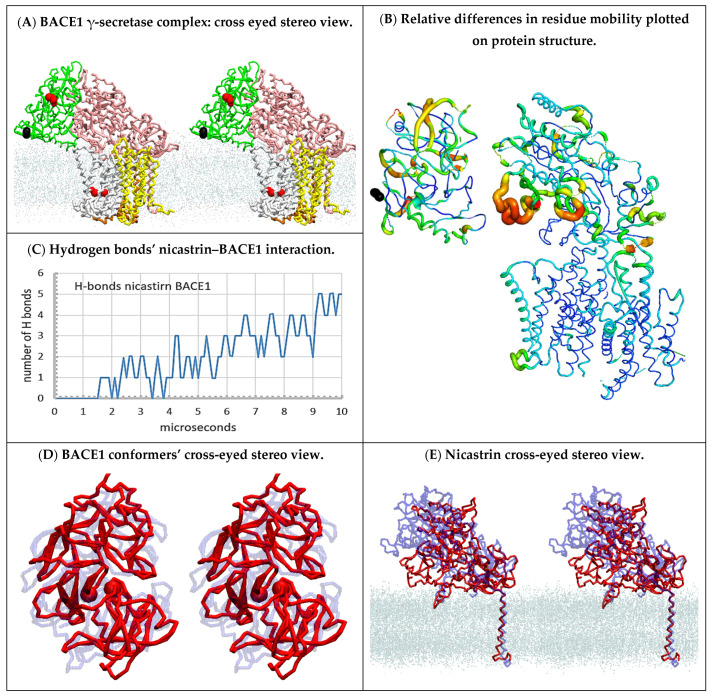
(**A**–**E**) Multiscale MD studies of supramolecular interaction between human BACE1 (PDB:4FGX, [54]) and γ-secretase (PDB:6IYC, [46]): (**A**) Cross-eyed stereo view showing a possible supramolecular complex between human BACE1 and γ-secretase (Appendix A). BACE1 (green) is shown with highlighted active site aspartates (red) and C-terminal domain (black). γ-Secretase is shown as nicastrin (pink), presenilin (silver), and Aph1 (yellow), while the active sites Asp 257 and Asp 385 are highlighted as red spheres. The position of the cholesterol–lipid bilayer is indicated with dots. BACE1 can spontaneously form a large docking surface with γ-secretase while its C-terminal end (black) is facing the membrane surface just where its transmembrane domain should start [54]. (**B**) The mobility of different protein parts during the complex formation can be illustrated using principal component analysis in Bio3D protocols [45]. The two proteins were positioned in the same orientation as in panel A, except that for clarity the complex was slightly spread apart, and the cholesterol–lipid bilayer is not shown. The thin blue lines represent the lowest mobility, green and yellow lines represent intermediate mobility, and thick red lines represent the highest mobility. For orientation, the C-terminal domain of BACE1 is shown as black beads. BACE1 docking affects different parts in the entire BACE1 structure and leads to the opening of the nicastrin ectodomain and TM2 on presenilin 1. (**C**) The rate of complex formation in MD calculations can be monitored by counting the H bonds formed between the proteins as a function of a calculated molecular time (Appendix A). The initial lag represents diffusion before the first contact. The stepwise increase in the number of hydrogen bonds represents conformational changes that drive the complex formation. (**D**,**E**) Repeated docking showed that the biggest interaction surface and the largest number of hydrogen bonds can be seen when both the active site cleft of BACE1 (**D**) and the nicastrin ectodomain (**E**) are open. The conformational changes are illustrated by the overlapping BACE1 and nicastrin ectodomain structures before the complex formation (blue) and after the complex formation (red). The dots indicate the position of the cholesterol–lipid bilayer.

## 3. Discussion

We demonstrated the significance of the presented two-substrate mechanism by showing that this mechanism can address a wide range of pathogenic changes that have been observed in different studies of Alzheimer’s disease. Numerous studies have suggested that γ-secretase has a separate substrate docking site and active site [7,9,10,17,30,46,55]. Here we go a step further. We show that γ-secretase can bind two different substrate molecules in parallel—one at the docking site and one at the active site (Figure 3, Figure 4 and Figure 5). The physiological significance of the presented two-substrate mechanism can be summarized around five closely related observations. These five observations can summarize changes in γ-secretase activity in different studies of Alzheimer’s disease. Development of early diagnostic methods and effective drug design strategies depends on our ability to connect observations from different enzyme-based, cell-based, animal, and clinical studies of Alzheimer’s disease [3,5,7,9,10,32].

### 3.1. The Two-Substrate Mechanism and Pathogenic Changes in the Types of Aβ Products

The two most frequently analyzed pathogenic events are an increase in the Aβ (x-42)/Aβ (x-40) ratio [13,48] and increase in the production of the longer, more hydrophobic Aβ products [10,13,48]. The presented two-substrate mechanism can explain the observed pathogenic changes in Aβ production (Figure 9).

A wide range of different studies have shown that gradual saturation of γ-secretase with its C99-βCTF-APP substrate leads to an increase in the Aβ (x-42)/Aβ (x-40) ratio and accumulation of the longer, more hydrophobic Aβ products [10,13,14]. From textbook enzymology, we know that a gradual increase in the saturation of γ-secretase with its C99-βCTF-APP substrate can lead to a gradual increase in the chances that γ-secretase can be exposed to two substrate molecules in parallel [18,19,20,21]. Briefly, the catalytic cycle of γ-secretase consists of three steps: substrate recognition and binding, catalysis, and product release (i.e., E+S→ES→EP→E+P [18,19,20,21]). Substrate recognition and binding is a limiting step when the enzyme is sub-saturated with its substrate ((E+S→ES [18,19,20,21]). The catalysis and the product release become limiting steps when the enzyme is increasingly saturated with its substrate (ES→EP→E+P [18,19,20,21]). Thus, the gradual saturation leads to increased chances that the second substrate can challenge the docking site while the enzyme is still processing its first substrate. Binding of the second substrate can affect the catalytic activity, i.e., switching from an E+S→ES→EP mechanism to an E+S+S→ES+S→ ES+SES→ EP_1_+SEP_2_ mechanism (Figure 9).

The second substrate can affect the most dynamic sites in the presenilin structures that control processive catalysis [28,46]. Subtle conformational changes in these sites can induce a shift from the Aβ 49-46-43-40 path to the Aβ 48-45-42 path, along with a toxic increase in the Aβ (x-42)/Aβ (x-40) ratio [14,23,28,46,56]. The substrate bound at the docking site can produce partial inhibition or even entrapment of the longer Aβ catalytic intermediates (Figure 3, Figure 4, Figure 5, Figure 6 and Figure 7). The inhibition and the entrapment can further facilitate saturation with the substrate (Figure 9 [18]). Thus, the entire process can facilitate the pathogenic events with a positive feedback mechanism (Figure 9 [18]).

γ-Secretase is far from saturation with its substrate under healthy physiological conditions in cells [6,7,9,57]. For example, transfections with APP genes or disease-causing APPsw mutations can increase γ-secretase activity in cells by as much as 10–50-fold [53]! Such increase in γ-secretase activity is possible only if the enzyme is at least 10–50-fold below saturation under healthy conditions [19,58]. All enzymes in cells are far below saturation with their substrate [19,52,58,59]. Sub-saturated enzymes are a crucial mechanism in the control of cell physiology [19]. Sub-saturated enzymes can give the fastest and linear response to changes in metabolism [19,52,58,59]. Such conditions give the cells maximal control of the enzyme activity and metabolism [19,58,59]. Sub-saturated enzymes can favor metabolic regulation by substrate channeling and supramolecular organization [52,58,59].

The presented two-substrate mechanism suggests that Alzheimer’s disease can be described as γ-secretase “choking” on its sticky substrates (Figure 9). Decreases in the catalytic capacity of γ-secretase can be caused by decreases in the maximal turnover rates for γ-secretase [16,32,60,61,62,63,64,65,66,67], increases in C99-βCTF-APP metabolism [68,69,70,71,72], or a combination of those two events [7,14,65,73,74,75]. We propose that measurements of decreases in the catalytic capacity of γ-secretase could be used in future studies of the pathogenic changes in γ-secretase activity [7,32]. The easiest approach to measure decreases in the catalytic capacity of γ-secretase is to observe the changes in saturation with its substrate [18,19,20,21]. Studies of changes in the saturation of γ-secretase with its substrates can follow standard protocols [18,19,20,21], which do not be exist for the “enzyme-cooking” studies” [76].

**Figure 9 ijms-24-01835-f009:**
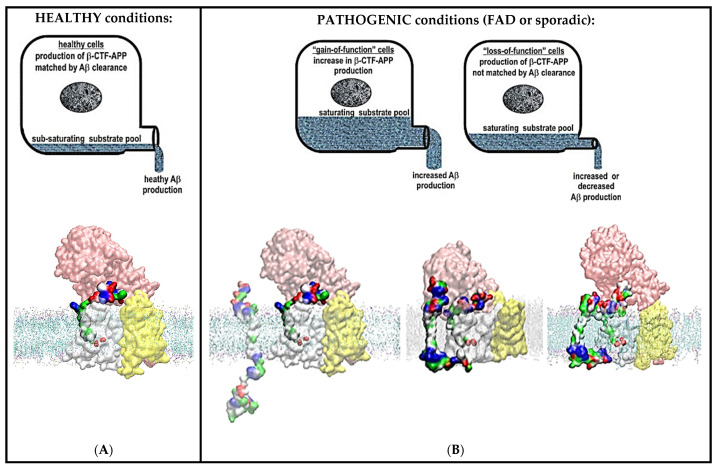
(**A**,**B**) Summary figure: decreases in catalytic capacity can trigger pathogenic changes in γ-secretase activity in sporadic and different familial cases of Alzheimer’s disease. The figure summarizes the physiological significance of the presented two-substrate mechanism. γ-Secretase’s function in cells can be illustrated as a drainpipe for cellular amyloid metabolism. Different levels of amyloid metabolism are illustrated as different levels of drain load. Alzheimer’s disease can be described at the molecular structural level as a mismatch between the optimal catalytic capacity of γ-secretase and amyloid metabolism, i.e., γ-secretase “choking” on its sticky substrate. (**A**) In healthy cells, the catalytic capacity of γ-secretase can match cellular levels of amyloid metabolism. γ-Secretase can completely process its different C99-βCTF-APP, C83-αCTF-APP, and Aβ substrates to soluble fragments with no interference [13,28,48]. (**B**) In pathogenic conditions, there is a mismatch between the optimal catalytic capacity of γ-secretase and amyloid metabolism. This mismatch can lead to an increase in the saturation of γ-secretase with its substrate due to an increase in amyloid metabolism (left), due to a decrease in the maximal activity of γ-secretase (right), or due to a combination of both effects. Any increase in the saturation of γ-secretase with its substrate can lead to an increase in the chances that the second substrate can challenge the docking site while the enzyme is still processing its first substrate [18,19,20,21]. The second substrate can bind to the most dynamic parts in the presenilin structures that can control processive catalysis and Aβ production [28,46]. The same sites can be affected by drugs and disease-causing mutations (Appendix A). The toxic interactions between the N-terminal domains of the two substrates are more likely with C99-βCTF-APP than with C83-αCTF-APP as the substrate (Figure 3, Figure 4, Figure 5 and Figure 6). This can explain why the β-secretase path is more pathogenic than the α-secretase path. Toxic aggregation between the N-terminal domains of the two substrates can be controlled by the closure of the nicastrin ectodomain (Figure 3, Figure 4, Figure 5 and Figure 6). Closure and opening of the nicastrin ectodomain can be controlled by supramolecular interaction between β-secretase and γ-secretase (Figure 8A).

### 3.2. The C99-βCTF-APP Substrate and Its Different Aβ Products Can Together Contribute to the Pathogenic Events

The presented two-substrate mechanism is the first mechanism that can explain the apparently conflicting observations that both the C99-βCTF-APP substrate and different Aβ products can lead to pathogenic events [36,67,72,77,78,79].

We propose that the toxic events start when the two substrates form contacts on the surface of γ-secretase (Figure 3, Figure 4 and Figure 5, Appendix A). The contacts can produce changes in γ-secretase’s structure (Figure 7) that lead to an increase in the Aβ (x-42)/Aβ (x-40) ratio, accumulation of the longer and more hydrophobic Aβ products, and accumulation of the C99-βCTF-APP substrate [32]. Any of these three events can lead to interference with the physiological functions of γ-secretase and, thus, a large number of cytotoxic events [80,81,82]. The initial complex between C99-βCTF-APP and Aβ fragments can seed aggregation with other C99-βCTF-APP or Aβ proteins that could ultimately lead to the formation of plaques [3,39,78]. The amyloid plaques are just the end result of the debris that gradually forms when γ-secretase is “choking” on its different products and the substrates (Figure 9 [3,39]).

The presented two-substrate mechanism does not exclude the possibility that even a third C99-βCTF-APP, C83-αCTF-APP, or Aβ molecule could bind to the presented two-substrate complex [9,10]. Such interactions could facilitate the entire aggregation process, which could ultimately lead to plaques [39]. The presented “choking” mechanism (Figure 9) can also trigger toxic interference with all other physiological functions of γ-secretase [1,2].

The presented “two-substrate-choking” mechanism could be an alternative to the proposals that toxic events start with the premature escape of hydrophobic Aβ proteins from the hydrophobic interior of γ-secretase [76]. The general assumption is that amyloid proteins ca be spontaneously released from γ-secretase to the lipid bilayer or even to an extracellular medium [3,39,78,83,84]. The released hydrophobic proteins start forming certain toxic Aβ oligomers by still-unknown mechanisms [39,85,86]. The spontaneous release of highly hydrophobic Aβ oligomers from the hydrophobic lipid bilayer is not consistent with basic biophysical principles [3,78]. The spontaneous release of highly hydrophobic Aβ oligomers from the hydrophobic lipid bilayer is not required to explain toxic events in the presented two-substrate mechanism (Figure 9).

### 3.3. The C99-βCTF-APP Path Is More Likely to Support the Toxic Two-Substrate Mechanism Than the C83-αCTF-APP Path

The presented two-substrate mechanism can explain why the C99-βCTF-APP path can be more toxic than the C83-αCTF-APP path [3,39]. Both C99-βCTF-APP and C83-αCTF-APP substrates can bind to γ-secretase while the enzyme is catalytically processing its different Aβ catalytic intermediates (Figure 3, Figure 4, Figure 5 and Figure 7). The longer N-terminal domains in C99-βCTF-APP substrates have larger interaction surfaces (Figure 3, Figure 4 and Figure 5). Thus, the C99-βCTF-APP substrate and the corresponding Aβ oligomers can produce greater interference with the physiological functions of γ-secretase and, thus, are more likely to lead to the related toxic steps [53].

### 3.4. The Two-Substrate Mechanism Can Explain Toxic Changes in Aβ Production in All Sporadic and FAD Cases of the Disease

FAD mutations are the best evidence that both increases and decreases in Aβ metabolism and γ-secretase activity can be observed in studies of pathogenic events [3,7,65,73,87]. Thus, the development of effective diagnostic and therapeutic approaches depends on our ability to understand factors that control the optimal γ-secretase activity and Aβ metabolism [3,16].

FAD mutations can act in parallel to other age-induced disruptions in optimal balance between γ-secretase activity and Aβ metabolism (Figure 9). The disruptions could be due to decreases in the maximal turnover rates for γ-secretase [16,32,60,61,62,63,64,65,66,67], an increase in C99-βCTF-APP metabolism [68,69,70,71,72], or a combination of the two events [7,14,65,73,74,75]. Such processes can be driven by changes in gene expression levels [64,88] or any other age-induced changes in cell physiology [16]. Thus, a large number of physiological processes can potentially support pathogenic changes in γ-secretase activity in Alzheimer’s disease [3,71,73,80]. Interestingly, any mismatch between the catalytic capacity of γ-secretase and APP metabolism can result in an increase in the saturation of γ-secretase with its substrate (Figure 9). Increases in the saturation of γ-secretase with its substrate have been observed in all of the studies of pathogenic events that have considered such a possibility [7,32,67,72]. The earliest age of onset can be observed with mutants that have the best chance to reach saturation at the lowest substrate loads [7]. The protective islandic A673T mutation in the APP substrate is the only mutation that leads to a decrease in γ-secretase’s saturation with its C99-βCTF-APP substrate [15]. The C-terminal domain of the second substrate and FAD mutations can affect the same presenilin structures (compare Appendix A). An increase in the saturation of γ-secretase with its substrate and different FAD mutations can induce shifts from the Aβ 49-46-43-40 path to the Aβ 48-45-42 path and support a toxic increase in the Aβ (x-42)/Aβ (x-40) ratio [10,13,14].

The presented two-substrate mechanism can explain how disruptions in the optimal balance between γ-secretase activity and Aβ metabolism can lead to toxic events in all different sporadic and FAD cases of the disease (Figure 9). A well-defined molecular mechanism that can connect different causes of the disease is crucial for the development of effective early diagnostic tools and drugs [19].

### 3.5. The Two-Substrate Mechanism and Development of Novel Drug Design Strategies

Drug development studies were among the first to indicate that γ-secretase has two substrate-binding sites [8,9,12,17,30]. Here, we show that the second substrate can affect the sites that bind different drugs (Appendix A). Such results are consistent with previous studies showing that a gradual increase in the saturation of γ-secretase with its substrate can affect how γ-secretase responds to drugs [6,7,9,11,12]. Drugs can lead to increases in the saturation of γ-secretase with its substrate [9,89]. Drugs, just like increasing the saturation of γ-secretase, can affect the Aβ (x-42)/Aβ (x-40) ratio [11,13]. FAD mutations can affect how drugs bind to γ-secretase [7,10]. Drugs, FAD mutations, and the second substrate can affect the most dynamic parts in the presenilin structure that control processive catalysis (Appendix A).

The presented insights indicate that the first possible improvement in future drug development strategies could be the development of competitive inhibitors of γ-secretase [19]. These competitive inhibitors could mimic the effects of the protective A673T mutation, i.e., decreasing the saturation of γ-secretase with its C99-βCTF-APP substrate [15,19]. Attempts to design competitive inhibitors that target the active site have been challenging. The attempts to target the active site with peptide analogs have been unsuccessful due to the surprisingly long and flexible active site tunnel [8,9,12,23]. Our results indicate three alternative strategies for the development of competitive inhibitors: First, the competitive inhibitors could be designed to facilitate the closure of the nicastrin ectodomain (Figure 3, Figure 4, Figure 5 and Figure 6). Second, the competitive inhibitors could compete with the formation of β-secretase–γ-secretase complexes (Figure 8). Third, the competitive inhibitors could be designed to bind to C99-βCTF-APP molecules and control its dimerization ([33,34,37] and Figure 3, Figure 4, Figure 5 and Figure 7). The development of compounds that target C99-βCTF-APP molecules is extremely difficult [39]. C99-βCTF-APP has a highly dynamic structure and, thus, represents a poorly defined target for effective drug development efforts (Figure 1 and Figure 2 [33,34,37]). The prepared drugs have to compete with other molecules that bind to C99-βCTF-APP with high affinity ([37] and Figure 1C).

The second major improvement in drug development strategies could be an expansion of the future target list. Compounds that can decrease the catalytic capacity of γ-secretase can be used to trace different physiological processes that control γ-secretase activity and amyloid metabolism at pre-symptomatic stages of the disease (Figure 9). Briefly, compounds such as semagacestat and avagacestat can be used in healthy animals to gradually induce pathogenesis, by provoking gradual saturation of γ-secretase with its substrate (Figure 9 [7,9]). The induced pathogenic events can be used for the description of physiological processes that control cellular levels of γ-secretase activity and/or total amyloid metabolism (Figure 9). Any physiological processes that control the balance between γ-secretase activity and total amyloid metabolism can be targets for future drug development efforts (Figure 9 [16,64,80,81,90]).

### 3.6. Concluding Remarks

The present study can be seen as an extension of earlier structural and docking studies [24,25,26,28,29,46]. However, the presented conclusions can be valid even if we do not know the precise substrate docking mechanism. Both C99-βCTF-APP and C83-αCTF-APP substrates can interact with γ-secretase while the enzyme is processing its different Aβ substrates (Figure 3, Figure 4 and Figure 5). The interactions can be transient contacts or a very specific complex. Any of those interactions can affect dynamic conformational changes that control processive catalysis by γ-secretase, but to varying extents (Appendix A).

The presented mechanisms support proposals that the majority of uncertainties and irreproducibility in studies of γ-secretase can be eliminated by controlling the saturation of γ-secretase with its substrate [6,9,11,12,13,18,19,91]. In all studies of γ-secretase activity we can observe competition between substrates binding to γ-secretase (Figure 3, Figure 4 and Figure 5) and substrates binding to the other substrates (Figure 2). In cell-based studies, dimerization of C99-βCTF-APP molecules is controlled by cell physiology, and it can grow out of control in cells that have non-physiological overexpression of C99-βCTF-APP molecules [6,91]. In enzyme-based studies, dimerization between C99-βCTF-APP molecules can be minimized by starting the assays with the substrates that come immediately after elution from the affinity column at low pH [10]. The measurements at different saturations require extra efforts and costs. However, such measurements can give consistent results and sustained progress in enzyme-based, cell-based, and drug development studies [7,8,9,10,11,13,48,91,92,93].

This study shows how multiscale MD studies can facilitate future studies of the molecular basis of Alzheimer’s disease and related drug development efforts [8,23,24,25,26,27,28]. The functional features of γ-secretase, C99-βCTF-APP, C83-αCTF-APP, and Aβ molecules depend on highly dynamic structures. Numerous transient contacts between flexible sites are driven by freely accessible charged, polar, and hydrophobic residues (Appendix A). Different functions of such dynamic and sticky molecules cannot be captured by static structural studies [23,33,39,46].

## 4. Materials and Methods

### 4.1. Preparation of Molecular Structures for Multiscale Molecular Dynamics (MD) Calculations

All MD studies started with full-length C99-βCTF-APP structures that were built from fragments of available NMR conformers (Val13 to Tyr58; numeration based on PDB: 2LP1 [33]). The missing parts in the NMR structures at the N-terminal end (residues 1 to 12) and the C-terminal end (residues 59 to 99) were built in several steps. First, the missing structures with no secondary structure presumptions were attached to the known transmembrane structures using Modeller 9.17 [94], i.e., as fully extended forms (Appendix A). Second, possible conformers in the cholesterol–lipid bilayer were calculated using multiscale MD studies and CHARMM-GUI tools [95]. Possible conformers were defined using coarse-grained MD studies that can depict as much as 20 µs of molecular events [38] (Figure 1 and Figure 2A–C, Appendix A) [95,96]. Finally, selected conformers and specific binding interactions were explored at the atomic level by converting selected coarse-grained structures to all-atom structures for all-atom MD calculations (Figure 2C) (AA-MD) [38,97]. All MD calculations started with the proteins that had a transmembrane section positioned in a cholesterol–lipid bilayer using OPM protocols [96].

Cryo-EM structures (PDB: 6IYC, [46]) can be used to prepare γ-secretase structures with Aβ substrates of different lengths in the active site tunnel. The missing loops in the γ-secretase structures were built with no secondary structure presumptions using Modeller 9.17 [94]. Modeller 9.17 can give between 5 and 10 conformers for short protein loops, which can be further optimized in multiscale MD protocols. Modeller 9.17 was further used to build missing parts in the N-terminal domain of the bound substrate. We prepared substrates with their N-terminal domain hidden by the nicastrin ectodomain to varying degrees. The substrates with their N-terminal domains placed in different positions were used for functional studies of the nicastrin ectodomain. Cryo-EM structures (PDB: 6IYC, [46]) could not capture the nicastrin ectodomain in a closed conformation, indicating that the function of the closed ectodomain depends on multiple conformers. MD calculations used γ-secretase structures with (PDB: 6IYC, [46], total 1355 residues) and without the bound substrate (PDB: 5FN2, [98], total 1309 residues). The 6IYC and 5FN2 structures showed differences between the structures with the active site tunnel in an open and closed conformation.

Structures of human BACE1 molecules (PDB: 4FGX, [54]) were used without inhibitors [52]. The missing 5-amino-acid-long loops were prepared using the Modeller 9.17 tool in UCSF Chimera [94].

Proteins were placed in a cholesterol–lipid bilayer that can define catalytically relevant presenilin structures [99,100]. The cholesterol–lipid bilayer can affect the relative distance and orientation between the active sites Asp257 and Asp385 (Figure 7B,C and Appendix A [99]). The relative distance and orientation can define the catalytic function of the active site aspartates—most notably pKa values [28,101]. The pKa calculations used MD structures that had the γ-carbon atoms on active site aspartates less than 4 Å apart [102,103]. Two different protocols were used to calculate the pKa values. The PropKa calculations for Asp257 and Asp385 were 6.9 and 6.8, respectively [102]. The Delphi calculations for Asp257 and Asp385 were 6.9 and 6.7, respectively [104]. These values are compatible with experimental observations showing that γ-secretase has optimal activity close to pH = 7.0 [103]. We found that homogeneous POPC bilayers that have been frequently used can produce some artifacts. Homogeneous POPC membranes cause an artificial charge distribution on the bilayer surface that can affect the structures of γ-secretase and its substrate. POPC membranes have also a loose packing that cannot support catalytically optimal presenilin structures ([101], Figure 7). Studies of γ-secretase activity showed that a specific mixture of CHAPSO (cholesterol), POPE, and POPC is crucial for the enzyme’s activity [10].

### 4.2. Coarse-Grained Molecular Dynamics Calculations

Coarse-grained (CG) MD calculations with γ-secretase and its substrates used the MARTINI 2.2 force field [105]. The CHARMM-GUI protocol [50] was used to prepare a simulation box with γ-secretase positioned in a cholesterol–lipid bilayer using the outputs from OPM protocols [96]. The smallest prepared box was 315 Å × 315 Å × 396 Å. The mixed lipid bilayer had 1612 lipid molecules, 106,668 water molecules, 1384 Na+ ions, and 1278 Cl− ions in a box. Periodic boundary conditions were employed in all directions, first with NVT and second with NPT boundaries applied. The cholesterol–lipid bilayer was assembled as follows: phosphatidylcholine (POPC), 340 molecules (21%); phosphatidylethanolamine (POPE), 176 molecules (11%); phosphatidic acid (POPA), 16 molecules (1%); phosphatidylserine (POPS), 64 molecules (4%); sphingomyelin (PSM), 96 molecules (6%); phosphatidylinositol (POPI), 32 molecules (2%); cholesterol (CHOL), 880 molecules (55%).

The prepared simulation box with the protein in the bilayer was subjected to two rounds of minimization, with the integrator set to *steep* and the number of integration steps set to *5000 or until default values have been achieved*. Four equilibration rounds came next, with the integrator set to *md*, and with the time step gradually increasing (5, 10, 15, and 20 femtoseconds). For all minimization and equilibration steps, the pressure coupling was set to *Berendsen, semiisotropic*, with *tau_p* set to 5.0 and compressibility set to = 3 × 10^−4^. The cutoff scheme was set to *Verlet*, the ns_type was set to *grid* with Verlet-buffer-tolerance = 0.005, and epsilon_r was set to 15. The Coulomb type was set to reaction-field, rcoulomb = 1.1, vdw_type = cutoff, vdw-modifier, Potential-shift-Verlet, rvdw = 1.1. Tcoupl = v-rescale, tc-grps = protein membrane solute, tau_t = 1.0 1.0 1.0, ref_t = 303.15 K.

The MD calculations used between 0.5 and 1 billion integration steps, with the integration time set to 20 femtoseconds. The calculation results were recorded in 1000 to 2000 frames, to depict a total of 10 to 20 µs of molecular events. The pressure (1 atm) and temperature (300 K) were held constant using a Langevin thermostat with a collision frequency of 1 ps-1. Bonds with hydrogen atoms were constrained using the SHAKE algorithm, while the long-range electrostatic interactions were calculated using the particle mesh Ewald method. The calculations used between 20 and 40 nodes on and Atos Bullx DLC 720 system and took about 3–5 days. Each node had two Xeon E5-2690 12C 2.6GHz processors (24 physical cores per node) and 64 GB RAM.

### 4.3. All-Atom Molecular Dynamics Calculations

All-atom molecular dynamics calculations (AA-MD) with γ-secretase and/or its substrates positioned in the cholesterol–lipid bilayer were prepared using the CHARMM-GUI Membrane Builder with the CHARMM36a force field [95,97]. Proteins were positioned in a lipid bilayer using OPM protocols [96]. OPM structures were placed in a typical water box with 1355 residues, 708 lipid molecules, 148,692 TIP3 water molecules, 419 Na+ ions, and 414 Cl− ions, in a 153 Å × 153 Å × 247 Å box (150 mM NaCl). The cholesterol–lipid bilayer was prepared as follows: phosphatidylcholine (POPC), 152 molecules (21%); phosphatidylethanolamine (POPE), 78 molecules (11%); phosphatidic acid (POPA), 8 molecules (1%); phosphatidylserine (POPS), 28 molecules (4%); sphingomyelin (PSM), 42 molecules (6%); phosphatidylinositol (POPI), 14 molecules (2%); cholesterol (CHOL) 386 molecules (55%).

The prepared simulation box was subjected to minimization, where integrator = steep, emtol = 1000.0, nsteps = 5000, nstlist = 10, cutoff-scheme = Verlet, rlist = 1.2, vdwtype = Cut-off, vdw-modifier = Force-switch, rvdw_switch = 1.0, rvdw = 1.2, coulombtype = pme, rcoulomb = 1.2, minimized in 5000 steps. The system was subsequently relaxed in 6 equilibration steps with the gradually increasing integration time. The setup for the equilibration steps was as follows: integrator = md, cutoff-scheme = Verlet, nstlist = 20, rlist = 1.2, coulombtype = pme, rcoulomb = 1.2, vdwtype = Cut-off, vdw-modifier = Force-switch, rvdw_switch = 1.0, rvdw = 1.2, tcoupl = berendse+n, tc_grps = PROT MEMB SOL_ION, tau_t = 1.0, ref_t = 303.15, pcoupl = berendsen, pcoupltype = semiisotropic, tau_p = 5.0, compressibility = 4.5 × 10^−5^, ref_p = 1.0, constraints = h-bonds, constraint_algorithm = LINCS, continuation = yes, comm_grps = PROT MEMB SOL_ION, refcoord_scaling = com.

The MD calculations took 3 to 6 days on an Atos Bullx DLC 720 system, using 20 to 50 nodes. Each node had two Xeon E5-2690v3 12C 2.6 GHz processors with 24 physical cores. The MD calculations used a system with the temperature set to 303.15 K, Nose–Hoover coupling, and the pressure set to 1.0 bar using semi-isotropic Parrinello–Rahman coupling. The calculations took between 100 and 200 million steps, with the step size set to 2 femtoseconds. The results were recorded in 150 to 200 frames, to depict between 200 and 400 nanoseconds of molecular events.

### 4.4. Statistical Analysis of Molecular Dynamics Results

The dynamic changes in molecular structures that can be observed in different molecular dynamics calculations can be quantified using statistical analysis with Bio3D protocols in the R 3.6.2 program [40]. The total degree of structural change that can be achieved in different calculations can be described by looking at converging RMSD values as a function of the molecular time (root-mean-square deviation). The differences in conformational changes at different molecular parts can be analyzed by looking at the RMSF values for each residue (root-mean-square fluctuation). The calculated differences can be also mapped directly on the protein structure by using principal component analysis.

### 4.5. Multiscale Molecular Dynamics Studies of Protein–Protein Interaction

Multiscale molecular dynamics studies of protein–protein docking are an iterative process that takes place in a sequence of complementary steps [38,47]. All docking studies started with the two proteins placed in different orientations and 5 to 30 Å apart. Different orientations can show how docking can be affected by the initial contact sites. Different initial separation distances can show how docking can be affected by diffusion and conformational changes that take place prior to complex formation (Appendix A). Coarse-grained MD studies can show how conformational changes can affect complex formation [47,105]. Selected structures from coarse-grained calculations were subjected to a more detailed structural analysis by using conversions from coarse-grained to all-atom structures [50]. The prepared all-atom structures can calculate the interactions down to each atom [50,95,97]. The results from different calculations were iteratively compared and correlated with the available literature. Docking studies were gradually optimized in the attempt to find conformers and contact sites that make maximal contact surfaces (Appendix A).

All docking studies with membrane-embedded proteins are largely affected by the position of each protein within the two-dimensional lipid bilayer (Appendix A). Soluble fragments of BACE1 molecules were the only proteins that were not embedded in the membrane [54]. In all docking studies with BACE1 molecules there is competition between free diffusion to the surrounding solution and interaction with the nicastrin ectodomain. The competition with free diffusion makes docking studies with BACE1 more rigorous and further demonstrates the significance of the presented supramolecular complex (Appendix A). Furthermore, BACE1 needs to dock to the nicastrin ectodomain with its C-terminal domain oriented towards the membrane surface (Figure 8). The C-terminal domain is the start of the transmembrane helix [54].

We found in all cases that the rate of interaction buildup, along with the number and position of the interaction sites, can be affected by the closure of the nicastrin ectodomain. We found that in all cases the diffusion distances between the two molecules had a relatively small effect on the rate of complex formation (Appendix A). The complex formation was primarily affected by the molecular flexibility and by competition between intramolecular and intermolecular interactions (Appendix A). The relative orientation between interacting molecules can affect the initial contact sites and the rate of interaction buildup. We focused our attention on the search for conformers that could give the biggest contact surface (Appendix A).

The number of H bonds as a function of molecular time can illustrate the rate of complex formation and its different phases. The initial lag represents the initial diffusion and the first contacts. The gradual increase in the number of H bonds represents conformational changes that take place during complex formation. The H bonds can be counted by extracting interaction surfaces that form in different steps in the MD calculations.

## Data Availability

Data is contained within the article or Appendix A. The data presented in this study are available upon request at https://cnrm.uniri.hr/, accessed on 10 January 2023.

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
