# Peer review of "The Binding of Different Substrate Molecules at the Docking Site and the Active Site of γ-Secretase Can Trigger Toxic Events in Sporadic and Familial Alzheimer’s Disease"

_ijms, 2023, doi:10.3390/ijms24031835_

Round 1
Reviewer 1 Report
The present study provides new evidence that γ-secretase can simultaneously bind two different substrate molecules. It has been shown that this interaction, similar to FAD mutations and various drugs, can interfere with process catalysis and promote toxic changes in the production of Aβ-proteins.
The article is written in great detail, and the theoretical analysis carried out is very scrupulous.
The approach of the authors looks quite convincing and opens the way for further research into the consequences of disclosing the distance between active site Asp257 and Asp385.
Observations on the physiological significance of the presented two-substrate mechanism are presented very clearly and intelligibly.
There are a few design notes:
The drawings are of low quality and very small. There is no need to include all the figures in the main text of the article; some of them can be moved to the SI.
There is no need to constantly provide cross-references (which is very distracting when reading the text), it is enough to enter them once or twice in the text.
The text of the Results section is written in too much detail. Thus, can concentrate on the most important thing, for a wide range of readers. However, this does not detract from the merits of this manuscript, which can be published in the submitted journal.
Author Response
We are grateful to the reviewers and the journal editors for their valuable comments and suggestions. Please accept our humble gratitude for your encouraging comments and suggested insights. We did some general changes following the reviewer’s recommendations. Some specific responses are inserted in green color to make them easier to follow.
We agree with the reviewer that the manuscript can be smaller and simplified. The size of the manuscript and the majority of additions were driven by the earlier rounds of the review that wanted more data and a higher level of proof. In addition, we wanted to make the text understandable to people with a wide range of expertise. We feel that the biggest problem in the current research is poor communication between different disciplines.
The present study provides new evidence that γ-secretase can simultaneously bind two different substrate molecules. It has been shown that this interaction, similar to FAD mutations and various drugs, can interfere with process catalysis and promote toxic changes in the production of Aβ-proteins.
The article is written in great detail, and the theoretical analysis carried out is very scrupulous. We apologize for writing such detailed and extensive text. Our research experience is that poor communication between different disciplines is the biggest limitation in the current studies of Alzheimer’s disease and drug development efforts.
The approach of the authors looks quite convincing and opens the way for further research into the consequences of disclosing the distance between active site Asp257 and Asp385. The calculations presented in this manuscript started in 2019 when PDB:6IYC structure become available. Multiscale MD studies are a new technique in the studies of Alzheimer’s disease that gave us a large amount of insightful data that can move the entire filed and drug design efforts forward. It is hard to choose to ignore some data.
Observations on the physiological significance of the presented two-substrate mechanism are presented very clearly and intelligibly.
There are a few design notes:
The drawings are of low quality and very small. There is no need to include all the figures in the main text of the article; some of them can be moved to the SI.
Most of the figures have been prepared in higher resolution or simplified to a different degree.
There is no need to constantly provide cross-references (which is very distracting when reading the text), it is enough to enter them once or twice in the text.
References were cut out through the text. Shorter and simpler sentences are presented. We apologize for so many references. It was our hope that a broad range of references can attract a wide range of experts. We feel that the biggest problem in the current research is a poor communication between different disciplines.
The text of the Results section is written in too much detail. Thus, can concentrate on the most important thing, for a wide range of readers. However, this does not detract from the merits of this manuscript, which can be published in the submitted journal.
Into, results, discussion section and figure legends are cut to different degree to make them simpler and easier to follow. Some information was transferred to the supplement section.

Reviewer 2 Report
This study first investigated the effect of the binding of different substrate molecules at the docking site on the toxic events in Alzheimer’s disease. The authors found the dimerization of C99-βCTF-APP molecules in the cholesterol-lipid-bilayer. Besides, the authors further studied the saturation of γ-secretase with its C99-βCTF-APP substrate. Moreover, the author also investigated the nicastrin function in the γ-secretase complex with the exposed N-terminal end of the bound Aβ substrate. Considering the clinical significance and logical methodology, this study can be considered to be published. However, several concerns still need to be revised before publication.
Minor revision:
1. The figure legend is not informative enough. The authors should revise that.
2. The authors should summarize the abbreviation to increase readability.
3. In the discussion part, the authors should fully summarize the current evidence regarding the effect of the binding of different substrate molecules on neurodegenerative diseases and highlight the clinical significance of the findings of this study.
4. There are some grammatical errors that need to be revised
Author Response
We are grateful the reviewers and the journal editors for their valuable comments and suggestions. Please accept our humble gratitude for encouraging comments and suggested insights. Our responses are inserted in green color to make them easier to follow.
We agree with the reviewer that the manuscript can be smaller and simplified. The size of the manuscript and the majority of additions were driven by the earlier rounds of the review that wanted more data and a higher level of proof. In addition, we wanted to make the text understandable to people with a wide range of expertise. We feel that the biggest problem in the current research is poor communication between different disciplines.
The calculations presented in this manuscript started in 2019 when PDB:6IYC structure become available. Multiscale MD studies are a new technique in the studies of Alzheimer’s disease that gave us many insightful data that can move the entire field and drug design efforts forward. It is hard to choose to ignore some data.
This study first investigated the effect of the binding of different substrate molecules at the docking site on the toxic events in Alzheimer’s disease. The authors found the dimerization of C99-βCTF-APP molecules in the cholesterol-lipid-bilayer. Besides, the authors further studied the saturation of γ-secretase with its C99-βCTF-APP substrate. Moreover, the author also investigated the nicastrin function in the γ-secretase complex with the exposed N-terminal end of the bound Aβ substrate. Considering the clinical significance and logical methodology, this study can be considered to be published. However, several concerns still need to be revised before publication.
Minor revision:
- The figure legend is not informative enough. The authors should revise that. We revised almost all figures and figure legends. We made the legends simpler. We can do any specific change that the reviewer is asking.
- The authors should summarize the abbreviation to increase readability. We apologize for such oversight. A list of abbreviations is now presented at the end. We have avoided abbreviations as much as possible.
- In the discussion part, the authors should fully summarize the current evidence regarding the effect of the binding of different substrate molecules on neurodegenerative diseases and highlight the clinical significance of the findings of this study.
We are especially thankful that the reviewer has encouraged us to link our data with the clinical studies. Many people have argued that Alzheimer’s disease is too complex to try to explain the pathogenic events at the level of molecular mechanism. In several places in the text we cited that insights at the molecular mechanism level are crucial for drug development studies (lines 74-78, and discussion section).
Please notice that in physiological conditions γ-secretase is far from saturation with its substrate, just as all other enzymes (Line 579-578). Interference with the other substrates could be an artifact of the studies that use genetically modified cells and animals that have extremely high levels of amyloid metabolism.
Please notice that there are over a hundred physiological substrates of gamma-secretase. Different substrate molecules have to be different manuscripts. The strongest genetic evidence connects C99-βCTF-APP substrate with Alzheimer’s disease. C99-βCTF-APP is an especially unusual, flexible molecule, with polar Thr43 and Thr46 residues in its hydrophobic TM region. No other molecule has such a structure. We extrapolate insights from C99-βCTF-APP substrate to speculate about the other substrates.
- There are some grammatical errors that need to be revised. We went over the text and found grammatical errors.
